# Towards a Multi-Pixel Photon-to-Digital Converter for Time-Bin Quantum Key Distribution

**DOI:** 10.3390/s23073376

**Published:** 2023-03-23

**Authors:** Simon Carrier, Michel Labrecque-Dias, Ramy Tannous, Pascal Gendron, Frédéric Nolet, Nicolas Roy, Tommy Rossignol, Frédéric Vachon, Samuel Parent, Thomas Jennewein, Serge Charlebois, Jean-François Pratte

**Affiliations:** 1Département de Génie Électrique et de Génie Informatique, Université de Sherbrooke, Sherbrooke, QC J1K 2R1, Canada; 2Institute for Quantum Computing, Department of Physics & Astronomy, University of Waterloo, Waterloo, ON N2L 3G1, Canada

**Keywords:** photon-to-digital converter (PDC), quantum key distribution (QKD), single-photon avalanche diode (SPAD), time-to-digital converter (TDC), CMOS detector, free-space, time-bin encoding, QEYSSat, quantum internet, quantum cryptography

## Abstract

We present an integrated single-photon detection device custom designed for quantum key distribution (QKD) with time-bin encoded single photons. We implemented and demonstrated a prototype photon-to-digital converter (PDC) that integrates an 8 × 8 single-photon avalanche diode (SPAD) array with on-chip digital signal processing built in TSMC 65 nm CMOS. The prototype SPADs are used to validate the QKD functionalities with an array of time-to-digital converters (TDCs) to timestamp and process the photon detection events. The PDC uses window gating to reject noise counts and on-chip processing to sort the photon detections into respective time-bins. The PDC prototype achieved a 22.7 ps RMS timing resolution and demonstrated operation in a time-bin setup with 158 ps time-bins at an optical wavelength of 410 nm. This PDC can therefore be an important building block for a QKD receiver and enables compact and robust time-bin QKD systems with imaging detectors.

## 1. Introduction

Quantum key distribution (QKD) is a key generation and sharing protocol where the security relies on quantum properties instead of computational complexity of certain mathematical problems such as in classical cryptography [1,2]. Typically in QKD, the qubits are photons sent from the sender to the receiver via fiber-optical networks, free-space links of space-to-ground links. Multiple methods of encoding information in single optical qubits (photons) have been developed [3,4,5,6]. One of these is called “time-bin” in which the information is encoded in the time of arrival and phase of the photon, which necessitates high-precision single-photon detectors at the receiver [7].

Time-bin encoding is achieved with unbalanced Mach–Zehnder interferometers (MZI) at both the sender and receiver. These must be as identical and stable as possible to avoid any phase drifts. The sender MZI prepares the qubit in one of four states from the time and phase bases: early (E), late (L), early and late with a constructive phase difference (ϕ=0), and early and late with a destructive phase difference (ϕ=π). The MZI receiver will detect one of four photon states: early–early (EE), late–late (LL), early–late (EL), or late–early (LE), which are temporally separated according to MZI path asymmetry.

Time-bin encoding offers advantages over polarization encoding in that it is relatively immune to depolarization and polarization mode-dispersion [8,9] of the channel and does not require polarization frame alignment of the sender and receiver. It is important to note that reducing the time separation between time-bins offers closer E and L time-bins, which allows for a more compact and stable MZI. This is, in particular, attractive for free-space applications where the distortion of the spatial mode requires elaborate imaging interferometers [9], particularly challenging for applications under stringent volume and mass restrictions such as satellite or handheld QKD. Typically, the outputs of the single-photon detectors, such as a single-photon avalanche diode (SPAD) [10,11] or a superconducting nanowire single-photon detector (SNSPD) [12,13], are connected to a time-tagger, a specialized chronometer often implemented in an FPGA. However, these configurations present two challenges: they can be bulky due to cooling or device size and suffer from signal degradation, because the connection circuitry and cables could reduce timing resolution. Furthermore, a conventional QKD system is based on single-pixel single-photon detectors. However, array detectors could be very beneficial for the system performance as they can enhance the photon count rate, perform quantum signal tracking tasks, and improve resilience against blinding attacks [14].

To address these challenges, a photon-to-digital converter (PDC) prototype is developed, comprising a single-photon detector array coupled to tailored digital electronics to process information such as the timestamp of each photon using embedded time-to-digital converters (TDC). This configuration can be realized in a very compact form (a few square millimeters for the PDC chip), and the distance between the detector and time-tagger is very short (<1 mm).

The PDC has an 8 × 8 SPAD [15,16] array in addition to the electronics on-chip (see Figure 1). This implementation is suboptimal because the 65 nm SPADs, although having a good timing resolution (7.8 ps FWHM SPTR at 410 nm wavelength [17]), suffer from low photon detection efficiency (7% at 410 nm in this work) and high noise (680 kcps average per SPAD). A more optimized solution would be to use another specialized technology for the SPADs and integrate them in 3D (on top) of the 65 nm CMOS electronics [18], which is the goal of future iterations. Although the shortcomings of 2D integration and the 65 nm SPADs were acceptable to demonstrate the QKD functionalities in this prototype, this PDC was designed to allow a future 3D integration and already includes top-side bond pads.

In the QKD post-processing, the key exchange requires a comparison of the absolute timestamps of all received photon detections with the corresponding emission times, via communication over a classical channel. Our custom PDC design directly supports this exchange. Iterating on previous work [19], the proposed PDC includes three features specifically targeted at QKD: timing window generation, TDC gating, and custom on-chip post-processing. The timing window generation allows the PDC to create variable-length gating signals on-chip when an external trigger is raised. It also allows the TDC to timestamp events relative to the end of the window signal. TDC gating uses the window signal to reject events outside of a window of interest. The custom on-chip processing converts the TDC code to picosecond timestamps directly and categorizes the timestamp into its time-bin value (EE, LE/EL, or LL). These added features allow the receiver’s detector to directly output the time-bin value, which reduces the data-volume substantially, thereby simplifying the entire QKD post-processing. These three components will be explained in further details in the Section 2.1. This is followed by a presentation of the testing and data acquisition setups in the Section 2.2. The electronic and optical performance results are then presented in the Section 3. Finally, a discussion of the results with a comparison to previous publications.

## 2. Materials and Methods

### 2.1. Architecture

#### 2.1.1. PDC Structure

The PDC was designed in TSMC 65 nm LP (low power) CMOS technology, with the goal to have the SPAD array integrated in 3D (on top) of the electronics. The 65 nm technology was chosen because it is small enough that it permits the TDCs and other circuits to fit in the restricted volume beneath each of the SPADs for 3D integration. This meant taking into account the footprints of the 3D bonding pads for each pixel. The PDC can be broken down into four main components, seen in Figure 1.

A—Registers and post-processing: registers are used to control the operating modes and the calibration parameters. The PDC has a suite of operations it can do on the timestamps before they are sent out of the chip. These operations will be referred to as the on-chip post-processing. There are two post-processing options that were specifically made in this PDC for QKD: “QKD Relative Timestamp” and “ QKD Time-Bin”, as seen in Figure 2. The first outputs the timestamp relative to the end of the gating window. The second attributes a time-bin value to the event and inserts that value to the output. The behavior of these post-processing operations are influenced by the settings of the registers. Registers are automatically placed and routed in the available area on the chip while respecting timing constraints.

B—SPADs: Single-photon avalanche diodes are photodiodes that are reverse biased beyond their breakdown voltage [15,16,18]. Because of this high electric field, an incoming single photon causes an avalanche current, which is detected and stopped by the quenching circuit. As this metastable operation is similar to that of a Geiger counter tube, the SPADs are said to be operated in “Geiger-mode”. The PDC has a 65 nm SPAD 8 × 8 array. The SPADs have a circular 20 µm diameter photosensitive area [17]. The total pixel size is 52.5 µm × 52.5 µm with a pitch of 60 µm. The 8 × 8 array totals 489 µm × 492 µm.

C—TDCs and quenching circuits: There is one TDC for every four SPADs (in a 2 × 2 configuration). Each SPAD has a quenching circuit that controls the avalanche process and resets the SPAD to be triggered again. These quenching circuits are located near the TDCs. TDCs are connected to the quenching signals and determine the time of arrival of the photons. The silver-blue squares in this section are the connection pads for eventual 3D bonding, although not used in this prototype.

D—Input/output pads: These are the IO pads for wirebonding the PDC. The ones on the bottom of Figure 1 are for control, communication, and voltage supply. The pads on the right are for wirebonding external SPADs.

Figure 2 shows how the different elements interact with each other and how the data flow through the PDC.

#### 2.1.2. TDC Gating

The TDC gating circuit (Figure 3) determines if an event (QUENCH_*) occurs inside or outside the window of interest in order to remove photon events associated with noise. Because the transmitted photons are sent at a fixed rate, the window is set to be open when they are expected to arrive. Any event that is outside of this window of interest is associated with noise (SPAD noise or ambient light). If outside of the window, it sends a reset signal to the TDC (OUT_WND) so that no timestamp is associated with the event. To keep the jitter of the signal going to the TDC to a minimum, the number of components from the quenching signal (QUENCH_*) to the TDC input is restricted as much as possible. In order to achieve best timing performance, the TDC gating is placed in parallel rather than in line with the SPAD signal. The TDC gating will send a reset signal (OUT_WND) to the TDC if the signal falls outside the gating window. Figure 2 shows the TDC gating in a system view where it is connected to the TDCs and gating window.

The most critical component of the TDC gating circuit are the arbiters. They determine which of the two input signals (WND_RE and QUENCH_RE for the top arbiter and WND_FE and QUENCH_RE for the bottom arbiter in Figure 3) is raised first. If IN1 is raised before IN2 then nOUT1 goes low (due to active low logic) and nOUT2 remains high. This stays until a reset signal is sent to the arbiter. This can be an external reset of the system, the end of the event (QUENCH_FE), or an internal reset due to the NO_Q signal.

#### 2.1.3. Data Format and Post-Processing

The data coming out of the chip are packaged in a custom structure and serialized at 250 Mbps. There are multiple data output formats available on the ASIC. The longest one requires 64 bits per TDC, so all data formats were set to this length to simplify data output logic, even if the data volume to be transmitted does not require it.

There are different post-processing modes that can be selected in the chip. The “QKD time-bin” mode uses the TDC gating and on-chip processing to extract which time-bin the event is attributed to. Figure 4 illustrates this: first, the timestamp measured by the TDC is made relative to the end of the window. Next, with on-chip programmable boundaries, the timestamps is categorized into one of five time-bins. Bins 1, 2, and 3 are for the three time-bins (LL, EL/LE, and EE) of interest, whereas bins 0 and 4 will contain noise that does not belong to a time-bin in the window. These two extra bins (0 and 4) are to offer more flexibility for assigning time-bins without having to change the window during operation.

Depending on the which post-processing is selected, the output data change formats. For example, Figure 5 shows the format for the “Time Conversion” post-processing, and Figure 6 shows the format for the “QKD Time-Bin” post-processing.

Depending on the post-processing selected, the volume of outgoing data can be reduced while retaining the desired information. In Figure 6, the most valuable information is the *Bin* and *Window* fields (total 24 bits). The *Bin* field (3 bits) indicates in which time-bin the event was classified. Because the sender and receiver must be synchronized in order to match the sent and received photons, the *Window* field returns the ID of the window, in other words, if it was the *n*th window. This would act as a counter to keep the sender and receiver events synchronized, and in a QKD exchange could be as few as 21 bits (26 ms overflow time at 80 MHz rate), as the other fields (40 bits) could be removed (see Figure 7). This device therefore helps overcome the data connection and data processing speed limitation a conventional detector and time-tagging based receiver would encounter for operation of a 64 pixel SPAD array.

To limit afterpulsing, the SPAD deadtime is configured to around 80 ns and, due to the set resolution, the approximate TDC deadtime is at most 30 ns. However, as these deadtimes occur in parallel and the internal post-processing is done in pipeline, the PDC’s theoretical throughput is limited mainly by the serial data output. As noted previously, the communication operates at 250 MHz with 64 bit frames. This means that for every event, it takes 256 ns minimum before the data are serialized out of the PDC. Effectively, the communication imposes a limit on the detection rate of 3.9 MHz. If only the 24 bits in QKD were used, this would increase the upper bound to 10.4 MHz (96 ns period), with the proper frame size.

### 2.2. Methods and Testing Setups

#### 2.2.1. Testing Setup Structure

The testing setup is divided into three subsystems: the ZCU102, the adapter board, and the head board, as seen in Figure 8. The ZCU102 is a commercial board with a ZYNQ FPGA [20]. The adapter board was designed and assembled in-house with all the circuits needed for timing tests (clock generators, delays, buffers, etc.). The PDC is wirebonded on the head board and contains the buffers and voltage regulators.

There are three main time-sensitive signals that are fed into the PDC: the event trigger, the window trigger, and the 250 MHz clock. Because any jitter on these signals adds jitter to the final result, these signals have to be generated with as little jitter as possible on the printed circuit boards (PCB). The event trigger acts as the start trigger for the TDCs and is routed via an H-tree to all TDCs. It is possible to program the PDC to enable or disable each TDC individually with the registers. The window trigger acts as the start signal for the gating window that is generated on-chip and is also routed in an H-tree. If enabled, the TDCs ignore all events that occur outside of this gating window. The window length can be programmed from 200 to 7500 ps. Finally, the 250 MHz clock acts as the system clock and is the the default “stop” signal of the TDC. That is, by default, timestamps are relative to the system clock of the chip. This can be changed via PDC configuration to use the end of the gating window as the stop signal. In this case, the timestamps become relative to the end of the gating window.

#### 2.2.2. Optical Tests Setup

The optical test bench uses an ultrafast MaiTai Laser operating at 820 nm that feeds an optical parametric oscillator (OPO) that produces a beam at 410 nm by means of second-harmonic generation. The laser pulse width is <100 fs at an 80 MHz repetition rate (±1 MHz) [21]. The 410 nm beam is directed towards the PDC detector. In order to reduce the laser intensity to be a single photon, neutral density filters are added before reaching the PDC. The 820 nm beam is sent into a low jitter photodiode. The electric pulse generated by this photodiode acts as the time reference and is sent to the PDC.

In typical operation, to get an absolute timestamp, the stop signal of the TDCs is the system clock. However, because we want a relative timestamp to the laser pulse emission for the time-correlated optical tests, the stop signal is programmed to be the end of the gating window. Hence, the output timestamp is the time difference between the arrival of the photon and the end of the window. If TDC gating is enabled, the TDC will ignore all events outside of this window. The window trigger signal can be the output signal of a laser, or, typically, the signal of a low-jitter photodiode as shown in Figure 9. The adapter board allows for a variable delay to shift the start of the window until a laser pulse arrives within the window.

In order to validate the on-chip time-bin QKD functionalities, the optical setup comprises a Mach–Zehnder interferometer (MZI). Figure 10 presents a schematic view of the MZI used in the test. As discussed previously, in time-bin QKD, there is a MZI at both the sender and receiver and they must be as identical as possible. To achieve this in a simplified setup, a mirror (M5) is used to send the photons back through the same MZI to reach a detector placed at the input. Mirror M3 and M4 are mounted on a translation stage (50 mm range) to adjust the path length of the second arm and thus the time difference between the bins. Figure 11 shows the full setup with the MZI and detector installed.

#### 2.2.3. Data Acquisition

The data acquisition chain was tailored for the custom PDC, from wirebonding, PCB design, FPGA interfacing, and finally data processing on a computer in Python (see Figure 12). In particular, the PCB was designed specifically to minimize the jitter, aiming for less than 10 ps RMS.

## 3. Results

This results section is divided into two main sections (electronic and optical) and presents the cumulative jitter from TDCs to time-bin optical measurements. This is to illustrate how the array integration and component design choices impacted the final time-bin measurements. The electronics section presents the results using only the trigger signals from the testing boards presented in Section 2.2.1. The optical section presents the time-bin results done with the setup presented in Section 2.2.2.

### 3.1. Electronic TDC Performance

The TDC architecture used is similar to previous work. More details on the timestamping conversion from TDC raw data can be found in [19,22].

As an example, Figure 13 shows for TDC #0 (head #7) a jitter of 7.48 ps RMS when measured using the system clock as stop signal and a correlated trigger signal generated on the adapter board with a time-delay between these two signals swept from 0 to 4000 ps with steps of 1 ps. We can define the total jitter of the TDC system as Equation (Equation 1). Because the start and stop signals are correlated, it is not possible to separate the contributions of the start and stop jitter. This is why both are considered together and the measured jitter was <4.2 ps RMS (start is an external trigger and stop is the system clock). This means that the jitter of the TDC can be calculated to be ∼6.2 ps RMS. However, because in a real setting, the jitter of the system (${\mathrm{jitter}}_{external}$) does impact the total performance (${\mathrm{jitter}}_{total}$), the total jitter will be used in results and comparisons. The jitter breakdown of Equations (Equation 1)–(Equation 3) is to understand which components have the most impact and if there are any bottlenecks. Note that in Equations (Equation 1)–(Equation 3), ${\mathrm{jitter}}_{\mathrm{external}}^{2}$, ${\mathrm{jitter}}_{\mathrm{start}\mathrm{and}\mathrm{stop}}^{2}$, and ${\mathrm{jitter}}_{\mathrm{external}\mathrm{trigger}\mathrm{and}\mathrm{clock}}^{2}$ are all the same. *External* is the more generic case, which is essentially the *start and stop* signals, which, in turn, are the *external trigger and clock* in this case.
(1)$$\begin{array}{ccc} {\mathrm{jitter}}_{\mathrm{total}}^{2} & = & {\mathrm{jitter}}_{\mathrm{TDC}}^{2}+{\mathrm{jitter}}_{\mathrm{external}}^{2} \end{array}$$
(2)$$\begin{array}{ccc} \; & = & {\mathrm{jitter}}_{\mathrm{TDC}}^{2}+{\mathrm{jitter}}_{\mathrm{start}\;\mathrm{and}\;\mathrm{stop}}^{2} \end{array}$$
(3)$$\begin{array}{ccc} \; & = & {\mathrm{jitter}}_{\mathrm{TDC}}^{2}+{\mathrm{jitter}}_{\mathrm{external}\;\mathrm{trigger}\;\mathrm{and}\;\mathrm{clock}}^{2} \end{array}$$

Thus, from ${\mathrm{jitter}}_{total}$ measured at 7.48 ps and ${\mathrm{jitter}}_{external\;trigger\;and\;clock}$ measured at 4.2 ps, this gives a ${\mathrm{jitter}}_{TDC}$ of ∼6.2 ps. Using the window’s end as the stop signal adds jitter to the measurement, as shown in Figure 14. In fact, the measured ${\mathrm{jitter}}_{external}$ changes to <5.2 ps since the stop signal is now the window signal.

The total jitter is a sum of the contributions of the TDC, the start trigger, and the stop trigger. Equation (2) now can be changed as follows:(4)$$\begin{array}{cc} {\mathrm{jitter}}_{\mathrm{total}}^{2} & ={\mathrm{jitter}}_{\mathrm{TDC}}^{2}+{\mathrm{jitter}}_{\mathrm{window}\mathrm{circuit}}^{2}+{\mathrm{jitter}}_{\mathrm{ext}.\mathrm{start}\mathrm{trigger}\mathrm{and}\mathrm{ext}.\mathrm{stop}\mathrm{trigger}}^{2} \end{array}$$

Thus, from ${\mathrm{jitter}}_{total}$ measured at 10.48 ps and ${\mathrm{jitter}}_{ext.\;start\;trig.\;and\;ext.\;stop\;trig.}$ measured at 5.2 ps, this gives a ${\mathrm{jitter}}_{TDC}+{\mathrm{jitter}}_{window\;circuit}$ of ∼9.1 ps. The difference in performance for ${\mathrm{jitter}}_{TDC}$ between Figure 13 and Figure 14 corresponds to 6.2 ps and 9.1 ps, respectively. This indicates that the window circuit on-chip adds roughly $\sqrt{9.1^{2}-6.2^{2}}=6.7$ ps RMS to the jitter. As 6.2 ps and 6.7 ps are so close, the jitter is essentially doubled when using the window circuit.

Figure 15 illustrates the performance uniformity of the TDC array. Although much care was taken to make the TDCs identical and the arrays as uniform as possible, variations in the fabrication process, temperature fluctuations, circuit mismatch, and voltage fluctuations will influence the performance between TDCs. The resolution of the TDCs are tuned with a digital-to-analog converter (DAC). However, as each TDC is slightly different, and the applied voltage is the same for every TDC, some performance variations will occur. The average jitter is 8.4 ps RMS with a 2 ps standard deviation.

### 3.2. Time-Bin Measurements

We now report on the operation of the PDC in the “QKD Time-bin” mode using the setup of Figure 10. The following results are from the optical tests in which the three time-bins are measured with the MZI shown in Figure 10 at 410 nm wavelength.

In Figure 16, the three time-bins (late–late, early–late/late–early, and early–early) are categorized on-chip. They correspond to bins 1, 2, and 3 respectively. Because a late event gets a smaller timestamp measured with respect to the end of the gating window, late–late corresponds to bin 1 (refer to Figure 4). Bins 0 and 4 correspond to events that are outside the bounds and are filled with noise triggered events. The bounds between each bins are programmable on-chip. This processing reduces the timestamp information (22 bits) to bin value (3 bits), which maximizes the potential throughput.

### 3.3. Jitter Estimation for the SPAD and Quenching Circuit Chain

For all time-bin measurements, the start signal of the TDC was the photon arrival and the stop signal was the end of the window signal. The window trigger signal is generated either from another photodiode or from a signal from the laser. Thus, the jitter equation from Equation (2) changes to:(5)$$\begin{array}{cc} {\mathrm{jitter}}_{\mathrm{total}}^{2} & =\left[{\mathrm{jitter}}_{\mathrm{TDC}}^{2}+{\mathrm{jitter}}_{\mathrm{window}}^{2}\right]+\left[{\mathrm{jitter}}_{\mathrm{SPAD}+\mathrm{QC}}^{2}+{\mathrm{jitter}}_{\mathrm{laser}\mathrm{reference}\mathrm{photodiode}}^{2}\right] \end{array}$$(6)$$\begin{array}{cc} 22.7^{2} & =10.48^{2}+\left[20.1^{2}+1^{2}\right] \end{array}$$

The $jitter_{\mathrm{laser}\;\mathrm{reference}\;\mathrm{photodiode}}$ was measured to be 3 ps FWHM with the same setup (laser, photodiode, and oscilloscope) as [17], thus the jitter of the reference photodiode is approximated to <1 ps RMS. From the electrical tests, the jitter of the TDC + window is 10.48 ps RMS when including the jitter of the setup. Therefore, given the $jitter_{total}$ = 22.7 ps RMS from Figure 16, that leaves a jitter of ~20.1 ps RMS for the SPAD + quenching circuit (Equation (Equation 6)).

This is significantly higher than the 7.8 ps FWHM of our previous publication [17]. There are four factors that could explain this difference: (1) this PDC has an array of SPADs instead of a single SPAD channel, (2) the end of the generated window is used as the TDC stop signal, (3) low power 65 nm CMOS (LP) is used instead of general purpose 65 nm CMOS (GP), or (4) design changes to the SPAD layout.

In this PDC, there are 64 SPADs that trigger at an average rate of 680 kcps that generate electrical crosstalk and noise into the power rails or the substrate. Each TDC has two ring oscillators that also generate common-mode noise. In turn, this noise couples to the TDCs and increases their jitter. Jitter measurements were taken while one SPAD channel was activated (others disabled) to compare to when all SPAD channels are active. The jitter did degrade by 1–2 ps RMS in the latter case. Although this does not explain the large difference, it is a contributing factor.

During experiments, it was noticed that some TDC timestamp values would suffer from more jitter when using the window as the stop signal. Although Figure 14 shows the impact across all codes, the window might negatively impact certain timestamps that would affect measurements such as those in Figure 16, which do not average the jitter across the whole dynamic range.

Due to a higher threshold and less leakage, LP offers lower power consumption at the expense of speed. Although measures were taken to account for these differences, the impact of the slower speed (compared to GP) might have been bigger than anticipated through simulation.

In this PDC, the quenching circuits read the cathode of each SPAD (as opposed to the anode in [17]). Although the SPAD’s architecture was not modified significantly with respect to [17], the connection to the cathode of the SPAD changes the parasitic capacitance at the quenching circuit readout node. For example, one could expect the cathode-to-substrate capacitance to be higher than that of the anode. Such changes would certainly degrade the jitter of the SPAD + quenching circuit.

Because in this PDC we do not have access to the window end signal or the SPAD cathode, it is difficult to give a definitive answer on the source of the added jitter of the SPADs. Although this is still under investigation, the next revision of the PDC will include new unitary test structures in the hope of clearly identifying the cause.

## 4. Discussion

The performances of the TDC array are an improvement over previous iterations [19] from our team and offers additional functionalities such as TDC gating. In [19], TDC jitter performance degradation had been observed when many TDCs were operating simultaneously, injecting noise in the substrate and the power rail. To address this, this iteration made three modifications based on the recommendations of [19]. First, to reduce the common-mode noise, in this integrated circuit there is one TDC for four SPADs. Second, to decrease the mismatch between each TDC, in this version the size of the oscillators’ transistors were increased. Third, to equally control the oscillators of the TDCs, we implemented the control voltages of the current starved elements from the ring oscillators in a mesh configuration. These changes proved beneficial, as Figure 17 shows that even in an array configuration with all elements active, the performance of the jitter and resolution (LSB) variations are improved with respect to [19].

The new window gating functionality, however, adds more jitter to the system. This can be seen when comparing Figure 13 with Figure 14 where the jitter increases from 7.48 ps RMS to 10.48 ps RMS just by using the window as the stop signal instead of the system clock. Because the measured jitter of the external signals (the “event trigger” and “window trigger”) going into the PDC is ∼4 ps RMS, the PDC is nearing the limits set by the test bench. However, as bigger arrays (such as 64 × 64) are an objective, ways to reduce the jitter and width variations of the gating window will be explored. The window generator is programmed on-chip to add or remove standard cells delay blocks to control the width of the window. As these delay blocks are susceptible to variations due to temperature and fabrication, the final width of the window can vary. The window size fluctuations and the routing of the window signal could explain the increase jitter from Figure 13 and Figure 14 and the increased jitter of SPAD+QC compared to the previous publication [17].

The SPAD array heats the device when activated and that heat is not properly distributed across the device. As seen in Figure 1, there is a SPAD array (B) next to the top side of the integrated read out (C). As they are noisy SPADs, good for electronics and functionality testing, they will generate a hot spot above the read out array (C). This impacts thermal noise and mismatch among electronics pixels. In addition, the routing capacitance between each SPAD and quenching circuit is not equal, which has an impact on the signal slope (I=C×dV/dt), which has a direct impact on the timing jitter.

More investigation is ongoing to understand the extra jitter observed in the measurement and better design techniques. A solution is to use the rising edge of the window instead of the falling edge as the time reference. This could reduce jitter, as the rising edge comes from the signal, compared to the falling edge, which is decided by a series of delay blocks in the window generator.

With the window gating, time-bins with 158 ps difference at 410 nm was achieved. This time difference represents a roughly 5 cm difference between each arm of the MZI and opens doors to explore compact MZI setups. In terms of time-bin separation, this result is similar to other publications [23,24], but this PDC offers the timestamping done on-chip. This means that no bulky external timestamping equipment is required. These elements, in addition to operating at room temperature, makes this detector a promising candidate for QKD in situations that require small size and low power consumption, such as hand-held or satellite free-space for a QKD network [25,26].

As noted previously, the SPADs used in this PDC were implemented to provide realistic input to the quenching circuits and TDCs and validate the new QKD functionalities. They are a stopgap solution before the future 3D integrated SPADs that will offer better dark count and photon detection efficiency (PDE). The 65 nm SPADs used in this PDC have a high noise (680 kcps average per SPAD across the 8 × 8 array) and low PDE (7% at 410 nm). Thankfully, the gating window could be used to reduce their impact. However, the noise restricted how low the laser power could go before losing the signal in the noise floor. This meant that the laser was not operating at a single-photon regime when measuring the time-bins. Because the objective was to demonstrate the functionalities of the PDC and not the security, operating at the single photon level was not an objective of this study but can be implemented in future work. In addition, the measured outgoing event rate coming out of the PDC was around ∼360 KHz (2800 ns period) in the experiment of Figure 16. This is much lower than the limit set by the serial communication noted previously (3.9 MHz). We estimate that with the high noise count of the 64 SPAD array, low PDE, and the window gating of 2.5 ns width, the majority of events are rejected by the window gating, and the SPADs are too often in their deadtime. Thus, we are not reaching the upper bound of the event rate.

## 5. Conclusions

The photon-to-digital converter concept allows us to integrate the full detection chain and some signal processing within a single device. In this work, the PDC was designed and implemented as a QKD receiver. The good timing resolution and jitter allows for around 158 ps separation between time-bins while maintaining photon detection rates of several MHz. This translates to more compact MZIs that can be implemented in space-restricted systems and offer easier calibration between the sender and receiver MZI. In addition, TDC gating allows us to reduce noise by only processing events that occur within the window of time the qubit is expected to arrive at the receiver. Finally, custom processing on-chip (such as on-chip time-binning) offers the possibility to filter unwanted events and extract only the essential data, thus increasing throughput.

These very unique QKD capabilities were demonstrated for this PDC prototype. The 8 × 8 array of 65 nm SPADs was used to provide realistic input to the system and validate the QKD functionalities of the PDC. Future work includes implementing 3D integrated SPAD design with the PDC to enhance the SPAD performance and adding further on-chip processing capability such as image analysis. 

## Figures and Tables

**Figure 1 sensors-23-03376-f001:**
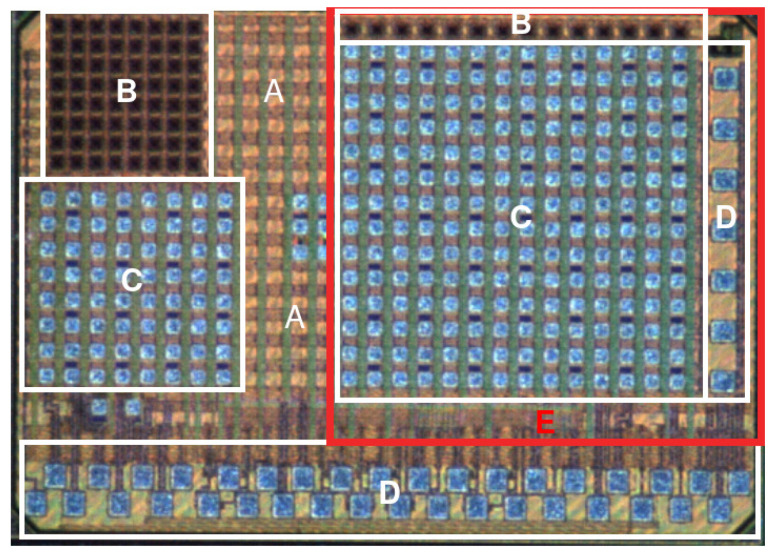
PDC diagram with the four main components: registers (**A**), SPADs (**B**), TDCs and quenching circuits (**C**), wirebonding pads (**D**). The rightmost array (red, **E**) was designed for another application and is not described in this paper.

**Figure 2 sensors-23-03376-f002:**
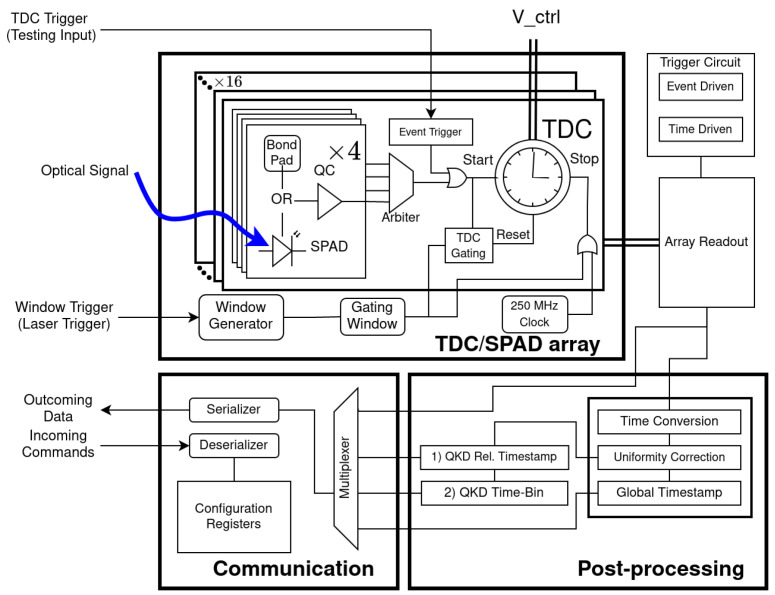
Block diagram of the all subsystems of the PDC, illustrating the signal flow. Each SPAD of the 8 × 8 array has its own quenching circuit, and 2 × 2 sub-arrays of SPADs are assigned to a TDC (16 in total). The array of TDCs is read out and the data pass through post-processing to the output serializer. The blue arrow indicates an incoming photon on the SPAD.

**Figure 3 sensors-23-03376-f003:**
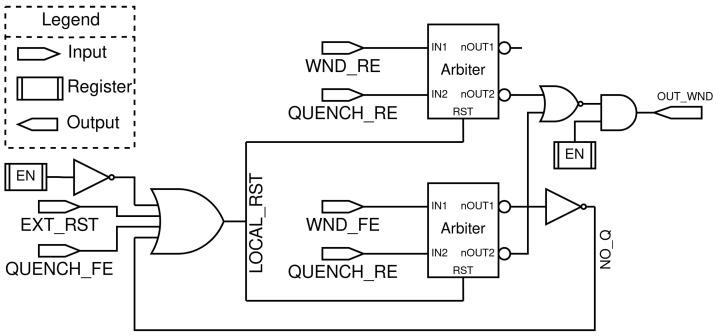
Simplified schematic of the TDC gating circuit that serves to identify three possible situations: (1) the QUENCH_RE is raised inside the window, (2) the QUENCH_RE is raised before the window, (3) there is no QUENCH_RE inside the window. Cases (1) and (3) are discarded by the circuit keeping OUT_WND low and self-resets. Case (2) causes the OUT_WND signal to raise and the TDC is maintained in a reset state until a next event is permitted. All *_FE (falling edge) and *_RE (rising edge) signals are created with D flip-flops with asynchronous clear.

**Figure 4 sensors-23-03376-f004:**
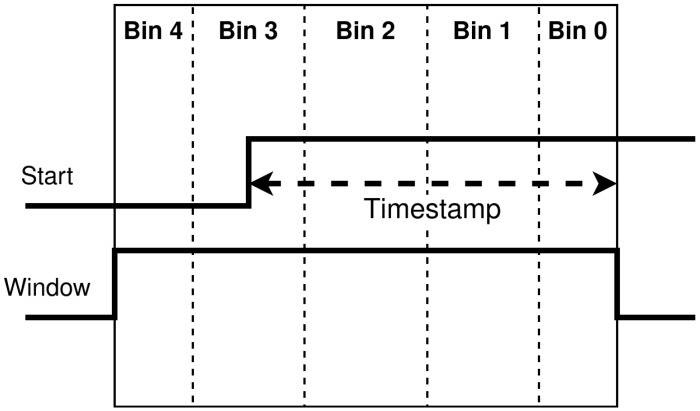
Using the TDC gating and programmable boundaries, on-chip processing can categorize events into which time-bin they belong. The timestamp is relative to the window instead of the system clock.

**Figure 5 sensors-23-03376-f005:**
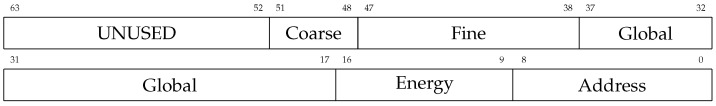
Dataframe of the the base output mode. Address [9 bits]: Address of the pixel. Energy [8 bits]: Number of hits received by that pixel since the last readout. Global [21 bits]: Timestamp of the system clock (250 MHz). Fine [10 bits]: TDC fine counter value. Coarse [4 bits]: TDC coarse counter value.

**Figure 6 sensors-23-03376-f006:**
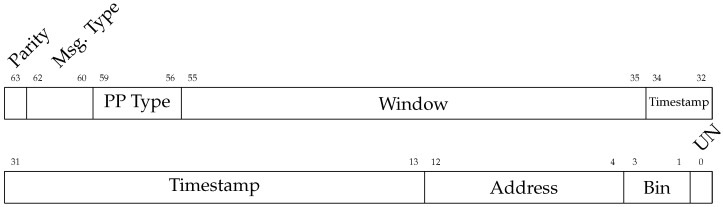
Dataframe of the the QKD output mode. UN [1 bit]: Unused. Bin [3 bits]: In which time-bin value the event is attributed to. Address [9 bits]: Address of the pixel. Timestamp [22 bits]: Relative timestamp to the end of the window in picoseconds. Window [21 bits]: Number of windows since last reset. PP Type [4 bits]: Post-processing type used. For example, “QKD Rel. Timestamp” or “QKD time-bin”. See Figure 2. Msg. Type [3 bits]: Message type. Indicates if it came from array the 8 × 8 or 1 × 14 array for example. Parity [1 bit]: Parity bit check.

**Figure 7 sensors-23-03376-f007:**
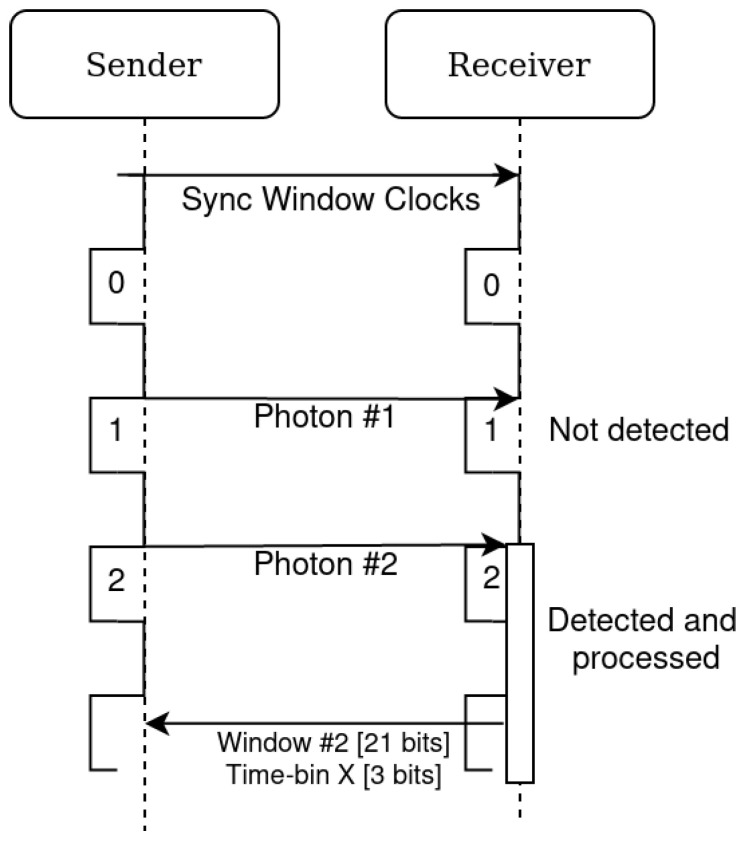
Diagram of the use of the *Window* field of the dataframe to synchronize the photons sent and received. The initial synchronization of the window clocks could be decided via a sequence of bright pulses or another absolute time reference. This synchronization of the window clock needs be done periodically to compensate for drifting.

**Figure 8 sensors-23-03376-f008:**
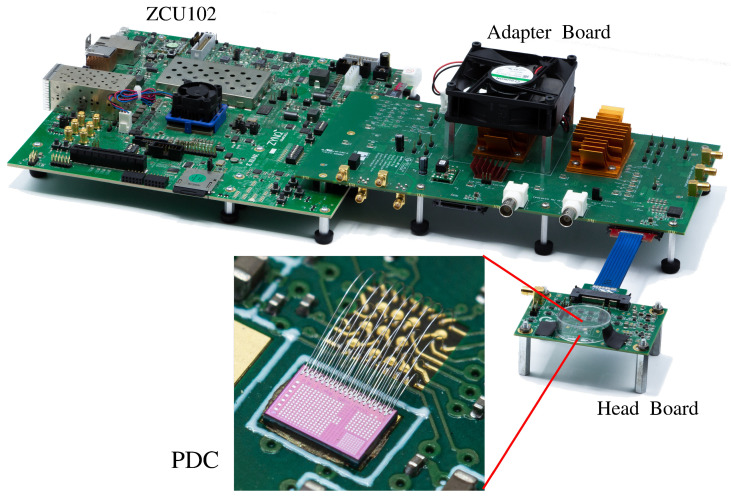
The complete electronic setup. The adapter board and head boards are connected via a SAMTEC cable to give flexibility to mount the head board of an optical setup. The ZCU102 and adapter boards are connected via a FMC connector to interface the critical signals with the FPGA. The PDC is wirebonded to the head PCB (zoomed view, bottom middle).

**Figure 9 sensors-23-03376-f009:**
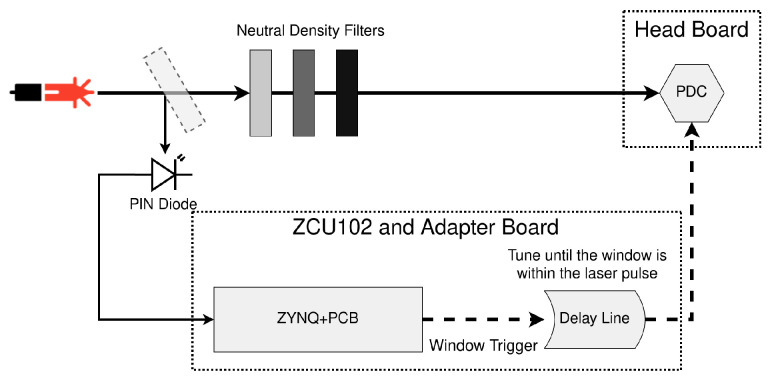
The setup used for the optical tests. The delay between the PCB and the head board of the window trigger signal is controlled on the board via the ZYNQ and Python scripts. The objective is to match the optical with this electronic delay so that the window trigger starts slightly before the arrival of the beam.

**Figure 10 sensors-23-03376-f010:**
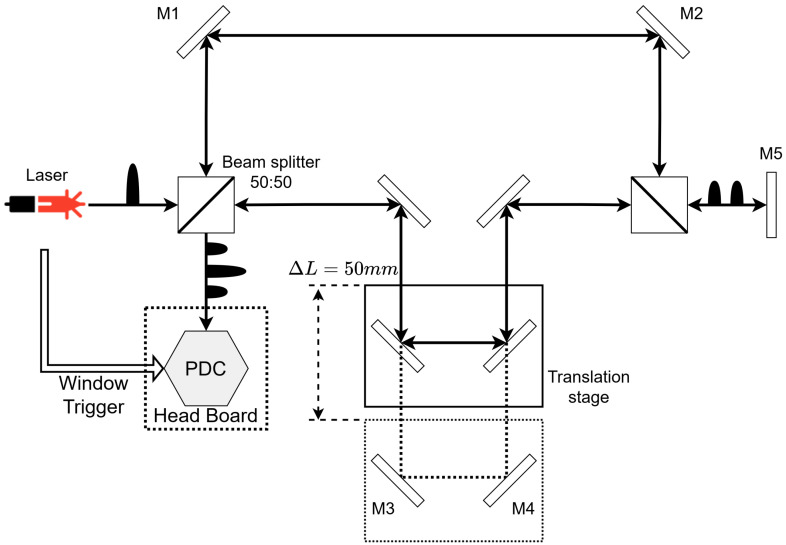
Basic MZI with a translation stage to control the time-bin separation. A mirror (M5) at the end makes the beam travel the MZI twice to mimic a full sender–receiver path.

**Figure 11 sensors-23-03376-f011:**
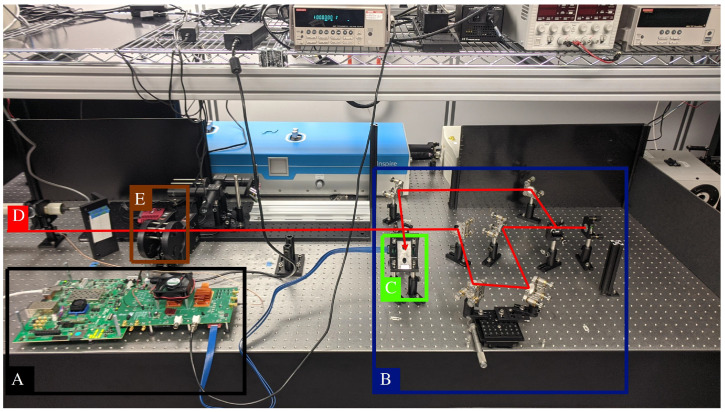
Laboratory setup for the Mach–Zehnder interferometer. The femtosecond laser comes in from the left (**D**, red). On the left, there is also the control board (**A**, black) for the detector from Figure 8. On the right is the MZI optical setup (**B**, blue) with the detector (**C**, green) in the middle, facing back. The optical setup (**B**, blue) is the same as the schematic of Figure 10. The neutral density filters (**E**, brown) can adjust laser power.

**Figure 12 sensors-23-03376-f012:**
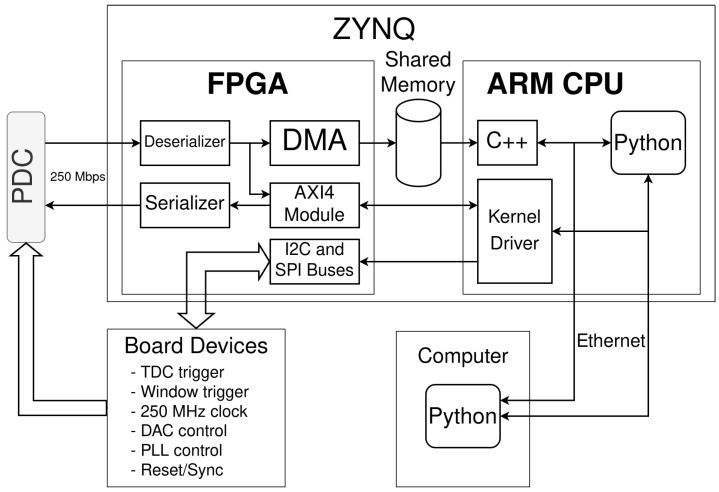
Simplified dataflow diagram for the data acquisition system. The direct memory access (DMA) allows one to send the data directly to memory space for the CPU to process. Python scripts then record, process, or control the PDC via the kernel driver. As the ARM CPU has access to all resources of the boards, the Python scripts automate most tests.

**Figure 13 sensors-23-03376-f013:**
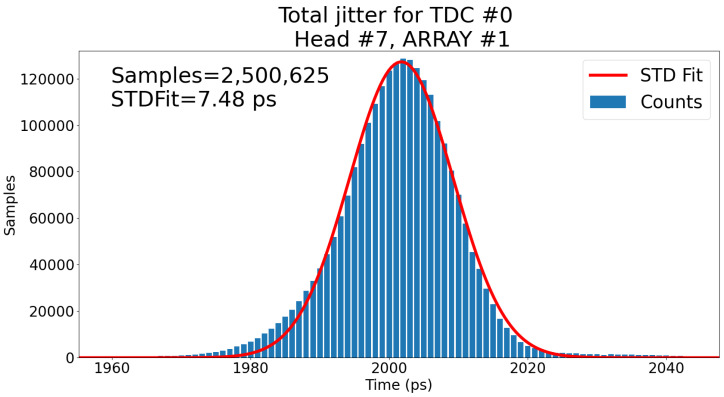
Total jitter of the TDC #0 of head #7 with all TDCs active for all codes. Sweep the clock-correlated start signal from 0 to 4000 ps with 1 ps steps, centered and aligned at 2000 ps. This result is the sum of all 4001 measurements and aligned. Even though the distribution is not Gaussian, the red line is a Gaussian fit to the whole distribution to obtain an estimate of 7.48 ps RMS for the jitter.

**Figure 14 sensors-23-03376-f014:**
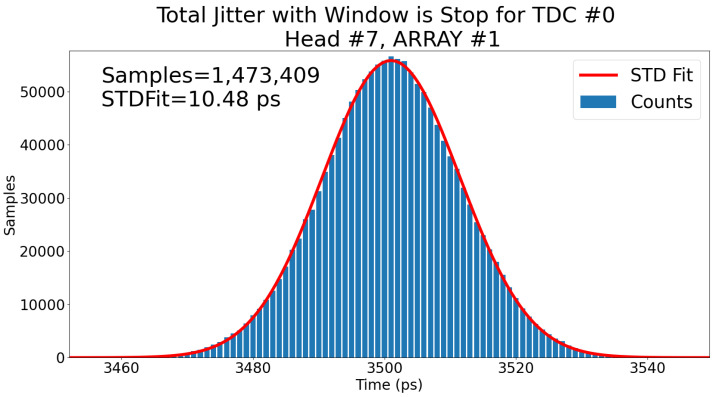
Total jitter of the TDC #0 of head #7 with all TDCs active with the window as the stop signal. The red line is a Gaussian fit to the whole distribution to obtain an estimate of 10.48 ps RMS for the jitter.

**Figure 15 sensors-23-03376-f015:**
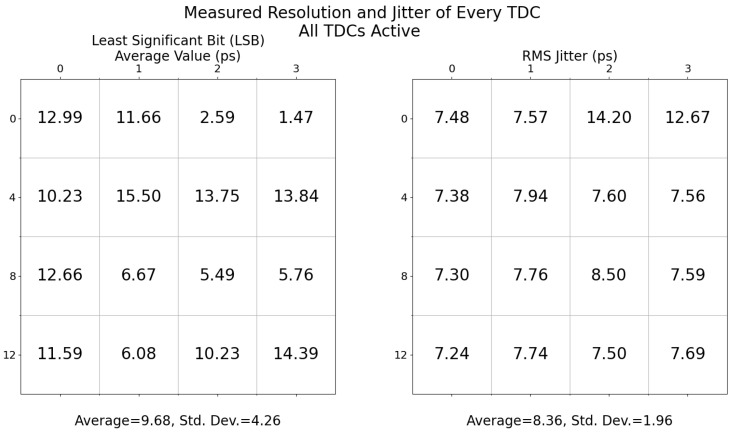
The resolution (LSB) and jitter of each TDC when all are operating at the same time. These results are for head #7 and array #1 of the chip. Due to variations in the fabrication process, not all TDCs have the exact same performance. This can be seen with the outliers, TDCs 2 and 3, having very fine resolutions and, consequently, higher jitter. The TDCs are indexed from the top left (0) to the bottom right (15), with the index of the leftmost indicated for each row.

**Figure 16 sensors-23-03376-f016:**
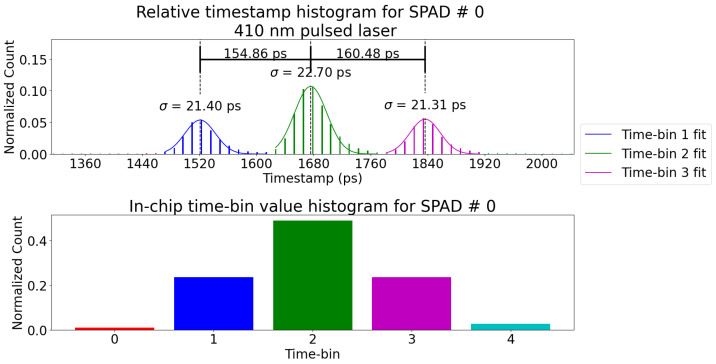
Time-bin histogram with on-chip timestamping and time-bin categorization. In this case, light was focused on SPAD #0 (connected to TDC #0) to compare the jitter with the previous results. The jitter increases from 10.48 ps to 22.7 ps RMS because the detection chain now includes the SPADs. Each event was categorized into a time-bin (0 to 4), which have programmable boundaries. The bottom histogram shows how many events were categorized in each bin, and the colors match bins between both graphs. Both histograms present the same information, either as relative time of detection or as on-chip categorized time-bins. Because the timestamps are relative to the end of the gating window, the late–late bin is #1, and the early–early is #3. The window size was set to 2.5 ns wide. The histograms are normalized so the total sum is 1.0.

**Figure 17 sensors-23-03376-f017:**
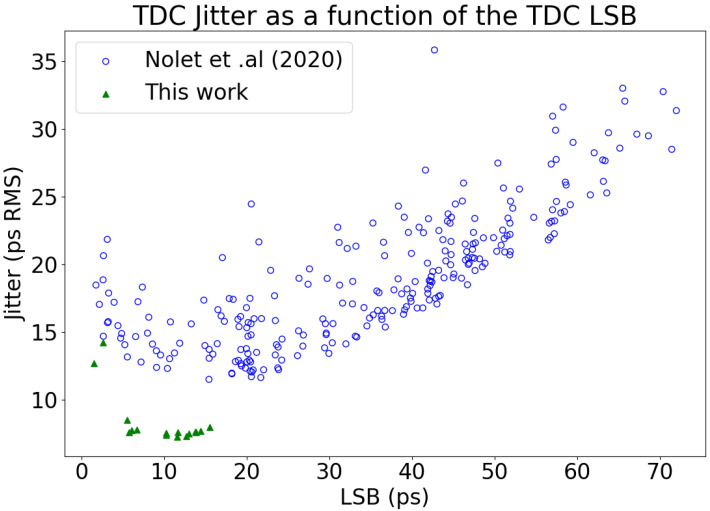
Jitter as a function of the LSB. The values from Figure 15 are compared to the results from Nolet (2020) [19] and illustrate the decreased variation of performance between TDCs. For example, in Nolet (2020), the LSB of every TDC would vary from 2 to 72 ps. In this work, this variation is from 2 to 16 ps.

## Data Availability

Not applicable.

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
