# Peer review of "Towards a Multi-Pixel Photon-to-Digital Converter for Time-Bin Quantum Key Distribution"

_sensors, 2023, doi:10.3390/s23073376_

Round 1
Reviewer 1 Report
Reviewer Comments (Sensors, MDPI)
This article addresses "Towards a Multi-Pixel Photon-to-Digital Converter for Time-Bin Quantum Key Distribution." The proposed scheme shows that the PDC was designed and implemented as a QKD receiver. The paper is well written.
- It is suggested to include more relevant references in the introduction section to find the differences between the prior research and the proposed scheme.
- The PDC has two 65 nm SPADs arrays: an 8x8 array. Why did the author particularly choose an 8x8 array? Does it affect the system process?
- The 5th paragraph of the introduction section is confusing with the reference details and the proposed one. Does the author mean the proposed scheme will work for 3D integration?
- Add the setup description for figure 10 in more detail.
- Figure 10, the caption should maintain the consistency for a, b, c, and d.
- In the result section it is mentioned that the electronics section presents the results only using trigger signals 219 from the testing boards presented in section 2.2.1. The optical section presents the time-bin 220 results done with the setup presented in section 2.2.2.
However, sections 2.2.1 and 2.2.2 present only setup details. There is no analysis of the results. So the style of writing seems irrelevant.
7. Signify the novelty of the proposed scheme.
- In the introduction, section clarify the section details.
- The conclusions drawn do not examine performance with sufficient analytical detail.
Reviewer 2 Report
As detailed in the attached word file.

Reviewer 3 Report
The proposed design looks very interesting since it is more cost competitive compared to other available designs by having more compact MZIs to be implemented in reduced spaces, however, I have some comments:
1) How is the KTC noise performances of the proposed PDC.
2) Please clearly explain the reason of performance degradation of the TDCs when running simultaneously. Is it due to power insufficiency, design problems…?
3) Please tag the laser path and all optical element used in the laboratory setup (figure 10). This will help better understand the setup.
Round 2
Reviewer 1 Report
The author has made clear all questions.